# Assessing Biosecurity Compliance in Poultry Farms: A Survey in a Densely Populated Poultry Area in North East Italy

**DOI:** 10.3390/ani12111409

**Published:** 2022-05-30

**Authors:** Giuditta Tilli, Andrea Laconi, Francesco Galuppo, Lapo Mughini-Gras, Alessandra Piccirillo

**Affiliations:** 1Department of Comparative Biomedicine and Food Science, University of Padua, Viale dell’Università 16, Legnaro, 35020 Padua, Italy; giuditta.tilli@phd.unipd.it (G.T.); andrea.laconi@unipd.it (A.L.); 2Unità Locale Socio-Sanitaria (ULSS) 6 Euganea, via Enrico degli Scrovegni 14, 35131 Padua, Italy; francesco.galuppo@aulss6.veneto.it; 3Centre for Infectious Disease Control, National Institute for Public Health and the Environment, 3720 BA Bilthoven, The Netherlands; l.mughinigras@uu.nl; 4Faculty of Veterinary Medicine, Institute for Risk Assessment Sciences, Utrecht University, 3584 CS Utrecht, The Netherlands

**Keywords:** biosecurity compliance, biosecurity checklist, broilers, turkeys, layers

## Abstract

**Simple Summary:**

In intensive animal farming, biosecurity represents a crucial point to guarantee animal welfare and health. Therefore, it is necessary that proper biosecurity measures are complied with and implemented. To date, several tools to assess biosecurity compliance in poultry farms are available with the general purpose of providing an evaluation of the implemented biosecurity measures. With this study, we aimed to assess the level of compliance with biosecurity measures in poultry farms located in the most densely populated Italian poultry area. Standardized biosecurity checklists according to national legislation were used to collect information. The main findings of our study show an overall good level of biosecurity implementation with some measures still requiring improvement. Our study may help to highlight the importance of biosecurity compliance in poultry farms by identifying the main achievements/challenges in a context in which measures are to be implemented by law and poultry production is vertically integrated.

**Abstract:**

Biosecurity in poultry farms represents the first line of defense against the entry and spread of pathogens that may have animal health, food safety, and economic consequences. The aim of this study was to assess biosecurity compliance in poultry farms located in a densely populated poultry area in North East Italy. A total of 259 poultry farms (i.e., broilers, turkeys, and layers) were surveyed between 2018 and 2019 using standardized checklists, and differences in biosecurity compliance between the poultry sectors and years (only for turkey farms) were tested for significance. Among the three sectors, turkey farms showed the highest compliance. Farm hygiene, infrastructure condition, cleaning and disinfection tools, and procedures were the biosecurity measures most complied with. Some deficiencies were observed in the cleanliness of the farm hygiene lock in broiler farms, as well as the presence of the house hygiene lock in broiler and layer farms and an adequate coverage of built-up litter in turkey and broiler farms. In conclusion, this study highlighted a generally high level of biosecurity in the visited poultry farms (probably due to the stringent national regulation and the integration of the poultry industry) and identified some measures that still need to be improved.

## 1. Introduction

In conventional poultry farming, the risk of transmission of infectious diseases, increased by, e.g., high stocking density, low genetic variation, suboptimal ventilation, and immunosuppression, represents a serious challenge for birds’ health and welfare [1,2]. Against this background, biosecurity represents one of the most powerful instruments to mitigate the risk of introduction (external biosecurity) and subsequent spread (internal biosecurity) of diseases between and within farms [3,4,5,6]. Proper implementation of external (e.g., entrance of visitors and vehicles, feed supply, and location of the farm) and internal (e.g., cleaning and disinfection, separation between poultry houses, and house hygiene lock) biosecurity should therefore be prioritized [3,6,7]. Moreover, the correct implementation of biosecurity may represent a successful intervention to reduce the risk of disease occurrence in poultry flocks and the need to administer antimicrobials that may lead to the emergence of antimicrobial resistance [8].

Biosecurity compliance is commonly assessed using questionnaires (or similar tools like checklists) in which the assessor answers a number of questions regarding the implemented biosecurity measures [7,9,10]. The final evaluation of both internal and external biosecurity can be either quantitative [7,11] or qualitative [9,12], according to the questionnaires and the national regulations in place. In the event of a negative outcome, different options can be applied, ranging from recommendations to penalties or even training of the farmer and personnel [13]. In previous publications, questionnaires or checklists proved to be useful tools for collecting data on biosecurity and assessing the compliance of poultry farms in the European Union (EU) [3,6,14] and in non-EU countries [11,12,15,16], highlighting potential deficiencies and opportunities for intervention.

In Italy, biosecurity measures in poultry farms are implemented following a national regulation according to which the farms must undergo periodic inspections by the official veterinary services that assess biosecurity compliance using nationally standardized checklists. Periodic and systematic biosecurity evaluation is required since Italy is among the largest poultry producers in Europe, accounting for approximately 10% and 12% of the total poultry meat and egg production, respectively [17,18]. The Italian poultry production is vertically integrated, with most of the broiler, turkey, and layer farms being located in the northern regions of the country (i.e., Lombardy, Veneto, and Emilia-Romagna). In particular, the Veneto and Lombardy regions have the highest number of poultry farms, accounting for 37% and 49% of national poultry farms and animals, respectively [19]. It follows that the implementation of biosecurity measures is crucial, since infectious diseases can severely affect the entire production chain. This was the case during the recurring avian influenza (AI) epidemics in Italy in the years following 1999 [20,21,22,23], which led to the establishment and the consequent enforcement of a national biosecurity legislation in 2005.

The objective of this study was to assess the biosecurity compliance in a sample of poultry farms located in North East Italy by analyzing data collected using nationally standardized checklists in 2018 and 2019. Specifically, the assessment aimed to identify potential deficiencies in biosecurity as well as to provide potential targets for intervention.

## 2. Materials and Methods

### 2.1. Poultry Farms

Data were collected from farms located in the province of Padua (Veneto region), North East Italy. Two hundred and fifty-nine farms were investigated over a two-year period: 188 in 2018 and 71 in 2019. Three sectors were considered: broilers (*n* = 126), turkeys (*n* = 111), and layers (*n* = 22). Farms were visited by the same official veterinarian as part of his duties once a year according to the epidemiological situation and regional recommendations at a frequency agreed with the local veterinary services. Although most of the farms were visited only once over the two-year period, some farms were visited twice. Specifically, three broiler farms, 48 turkey farms, and one layer farm were visited once in 2018 and once in 2019.

### 2.2. Biosecurity Checklists

Checklists for biosecurity compliance are frequently revised (at least annually); therefore, some questions are deleted or added each year. In this study, poultry sector-specific checklists were used that differed slightly between 2018 and 2019. Overall, the checklists for broiler and turkey farms comprised 182 questions divided into 19 sections, while the checklist for layer farms included 231 questions divided into 22 sections (Appendix A). Within each productive category, only questions included in both the 2018 and 2019 checklists were considered for data analysis. Each section aims to evaluate different categories of biosecurity measures: structural characteristics of the farm, employees (e.g., number, level of education and training), access control systems (i.e., parking or barriers), cleaning and disinfection procedures, bird management, litter and manure management, registers, and pest control. In addition, sections on egg management are included in the layer checklist. Most of the questions require a ‘yes/no’ answer, while a few are open questions. An official veterinarian from the local veterinary service fills in the checklist during the farm inspection, which usually occurs during the downtime between production cycles. Some questions are asked by the official veterinarian to the farmer during face-to-face interviews, and therefore rely on the trustworthiness of the latter. The outcome is considered favorable when the inspected farm does not show any non-compliance related to biosecurity. If the outcome is not favorable, corrective measures are reported in the checklist and recommendations or penalties are applied. Due to the large number of variables collected and analyzed in this study, only data related to the routes of transmission among and within farms (i.e., access to the farm, cleaning and disinfection procedures, and fresh and built-up litter) are reported and discussed, considering their role in hampering the introduction and spreading of pathogens within and between farms, respectively. The complete list of variables (50 variables for broilers and turkeys and 45 for layers) is included in the Appendix A.

### 2.3. Statistical Analysis

The statistical analysis aimed to evaluate the compliance with biosecurity measures in the three sectors (i.e., broilers, turkeys, and layers). Only the ‘yes/no’ questions present in both 2018 and 2019 checklists related to the routes of transmission (*n* = 50 and *n* = 45 for broilers and turkeys, and layers, respectively) were included in the analysis. Data were first presented descriptively as percentage of positive answers (i.e., ‘yes’) to each question related to the two-year period. Three different datasets (one per sector) were created, each referring to the two-year period. To investigate the level of implementation of biosecurity measures in each poultry sector, a 70% cut-off was adopted based on the (left-skewed) frequency distribution of biosecurity compliance among all farms, as 70% represents the overall mean value and upper threshold of the distribution tail after which highly compliant farms are located. A “high compliance” therefore refers to farms that score above the overall average. Fisher’s test was used to compare differences in biosecurity compliance (i.e., broilers vs. turkeys, broilers vs. layers, turkeys vs. layers) in the three sectors over the two-year period (i.e., 2018 and 2019). McNemar’s test was used to compare the evolution in biosecurity compliance in turkey farms from one year to the following. Differences were considered significant if the *p*-value was < 0.05. The statistical analyses were performed using GraphPad Prims v9.3.1 software (http://www.graphpad.com, accessed on 18 April 2022).

## 3. Results

In 2018, 188 biosecurity checklists were filled in, of which 30.9% were for turkey (58/188), 5.3% were for layer (10/188), and 63.8% were for broiler (120/188) farms (Table 1). Seventy-one checklists were completed in 2019, of which 74.6% were for turkey (53/71), 16.9% were for layer (12/71), and 8.5% were for broiler (6/71) farms (Table 1).

Most of the biosecurity measures showed a high level of compliance (>70%), particularly regarding external biosecurity (Figure 1).

Even though data showed a generally high level of implementation of biosecurity measures in term of both internal and external biosecurity in all poultry farms, some differences were observed among the three sectors, with turkey and broiler farms being the most and least compliant, respectively (Figure 1). In detail, broiler farms were slightly less compliant with a few measures such as the cleanliness of the farm hygiene lock premises. Similarly, they were less compliant with the implementation of the house hygiene lock, along with layer farms.

Biosecurity compliance in turkey farms showed an increase from 2018 to 2019. As reported in Figure 2, improvements were observed in the presence of the house hygiene lock; washable and disinfectable floors, walls, and roofs; and clean or disposable clothing for the personnel, combined with a decrease in the dirtiness of the equipment used for vehicle cleaning. Finally, although the absence of pumps for disinfection was rarely detected in turkeys in general (Figure 1a), the presence of pumps for disinfection decreased in occurrence between the two study years, indicating increased compliance with the enforced national law (Figure 2).

## 4. Discussion

This study aimed to assess the biosecurity implementation level using standardized national checklists in a sample of poultry farms located in one of the most densely populated poultry areas in Italy. Although data collection with questionnaires represents a snapshot of biosecurity implementation and the reliability of farmers’ answers to some questions might be arguable, questionnaires have proven to be a useful tool for assessing biosecurity compliance in poultry farms [3,6,11,15] and a potential driver for improvement even for the farmers themselves.

Our data showed a generally high level of implementation of internal and external biosecurity measures in all poultry farms. However, some differences were observed among the three sectors, with turkey and broiler farms being the most and least compliant, respectively. The high compliance in turkey farms might be due to the stricter national legislation implemented in this sector as compared to the others. Indeed, turkeys require longer rearing times and since AI appeared to affect turkeys more severely in the large 2017 outbreak [24] Veneto region established additional legislation to improve biosecurity compliance mainly among turkey farms. The increasing trend in biosecurity compliance observed between the two years in turkey farms may be the result of continuous efforts to assess biosecurity implementation (annual checks) and to amend legislation (annual revisions).

External biosecurity is fundamental to prevent the entry of pathogens into a poultry farm and our findings showed that external biosecurity was slightly more compliant than internal biosecurity. Specifically, the cleaning of ‘filter zones’ (which are comparable to a farm hygiene lock), having washable and disinfectable premises, and the presence of clean basins, cleaning equipment (i.e., liquid or bar soap, disposable or clean towels or hand dryers, and lockers for clothes), and clean footwear were found to be the most complied-with measures among those considered. This suggests that farmers are aware of the biological risk posed by people and do acknowledge the importance of cleaning and disinfection prior to entering a farm [25]. However, as also reported by Scott et al. [26], these findings should be interpreted cautiously, since the presence of, e.g., cleaning equipment, such as hand washing stations, showers, etc., does not necessarily imply their implementation. Indeed, some biosecurity practices rely on what the farmer declares when interviewed, since the official veterinarian cannot evaluate them directly at each moment. Other variables related to external biosecurity, such as access control (i.e., gate/bar closed on arrival), disinfection of vehicles (i.e., spray bay with a waterproof floor), and animal control, and those related to internal biosecurity, such as disinfection of house premises (i.e., intact walls, written procedure for cleaning and disinfection, and washable and disinfectable floors, walls, and roofs), showed high biosecurity compliance, thereby representing a landmark in biosecurity implementation. Indeed, proper cleaning and disinfection protocols are crucial to limit the spread of pathogens [27], while intact walls reduce the presence of invertebrates, which otherwise can hide in cracks and act as vehicles for poultry pathogens [28]. In addition, high percentages of no evidence of rats/mice was detected (even though they were lower for layer hens), highlighting the efficacy of the pest control procedures adopted by farmers.

Remarkably, attention to litter management (e.g., a platform with a waterproof floor for temporary storage of built-up litter and fresh litter used without storage) attributable to external biosecurity seems to be high, and this might be crucial for limiting the spread of pathogens among flocks and may also have potential environmental consequences. According to our findings, the direct use of fresh litter without storage is the most adopted technique in both broiler and turkey farms. This may represent a successful practice because if the fresh litter is stored prior to use it could attract pests and wild animals and contribute to the introduction of pathogens into the poultry houses. Temporary storage of built-up litter, if improperly managed, can lead to low air quality due to the release of carbon dioxide, methane, and ammonia [29]. Lastly, pumps for disinfection were not commonly found in layer farms. This could be easily misunderstood as a step back in biosecurity compliance. However, this represents a step forward since from 1 January 2020 Italian law has forbidden the use of pump systems for vehicle disinfection. Instead, the law requires the adoption of a fixed and automatized disinfection system (e.g., disinfection arch), which is more efficient and effective.

The strict biosecurity laws enforced since 2005 as a consequence of AI outbreaks [20,30], together with the organization of the poultry industry (vertical integration), might have contributed to the achievements highlighted in this study. The positive influence of integration in poultry production on biosecurity compliance has been previously reported by East [12] and may be explained by the fact that each integrated company usually has its own biosecurity policy in addition to the measures established by national biosecurity legislation.

The lack of compliance observed in broiler and layer farms with regard to some measures (e.g., house hygiene lock) could hamper the correct management of farm and poultry houses and represents a risk factor for the introduction and spread of infectious diseases [31]. In particular, a house hygiene lock at the entrance of the poultry house acts as an important barrier allowing for the separation of flocks [32,33], hence reducing the risk of pathogen spread. Therefore, more effort should be made to increase the use of house hygiene locks in both broiler and layer farms. Furthermore, the presence of clean footwear in the house hygiene lock should be improved in layer farms considering that the longer the rearing cycle the higher the risk of the introduction of infectious agents.

Notably, a high presence of water bodies near turkey farms was recorded. This might be an intervention point for external biosecurity to prioritize, since water bodies are a risk factor for the introduction of AI viruses due to the presence of waterfowls [24,34] and should be removed from farm surroundings [26]. However, this does not always seem to be a straightforward measure to implement.

In addition, adequate coverage of the built-up litter (manure) should be an important intervention target for external biosecurity in broiler and turkey farms as built-up litter might represent a reservoir for poultry pathogens [35]. Greater attention should also be paid to litter management during the rearing cycle (e.g., milling or addition of litter) in both turkey and broiler farms, since these practices involve entry into the poultry house and could represent a potential risk for the introduction of pathogens. Moreover, pumps for disinfection were commonly found in broiler farms. Even though this practice is not allowed by the national legislation in effect since January 2020, this represents an intervention point, and during a visit to broiler farms attention should be paid to the correct implementation of alternative tools for disinfection. Our findings also show potential intervention points in layer farms, such as the scarcity of a written procedure for cleaning and disinfection (internal biosecurity).

Interestingly, during the study period no relevant disease outbreaks were reported in the area [36,37] and high compliance with biosecurity measures might have contributed to this. Healthy animals do not need any (or little) treatment; therefore, coordinated and continuous attention to biosecurity measures over the years might have also contributed to decreased antimicrobial use (AMU) in Italian poultry farms [38,39]. It is worth highlighting that human and environmental health could also benefit from a reduction in AMU in poultry farms as antimicrobial residues and resistance could spread into the environment due to manure application to soil and may reach humans through water sources and/or via the food chain [40,41]. Future studies should aim to investigate the exact role of biosecurity in reducing AMU.

## 5. Conclusions

Biosecurity compliance represents a fundamental step for preventing the introduction and spread of pathogens in intensive poultry farms, and several biosecurity measures require continuous implementation and training of personnel. However, while data collection with questionnaires have proven to be a useful tool, in certain cases it reports a snapshot of the biosecurity endeavor representative of the moment at which the questionnaires are completed. This should be taken into account when interpreting the data since the presence or absence of a specific biosecurity measure at a given time may not correspond to the effective and constant implementation of the measure.

Our findings show a generally high level of implementation of biosecurity in poultry farms in North East Italy. This may be explained by several factors, including strict national biosecurity regulations and the widespread presence of integrated companies governing the poultry production system. Our study shows that there is still room for improvement with regard to some aspects of biosecurity compliance (both internal and external), such as the presence of clean footwear in house hygiene locks, the presence of written procedures for cleaning and disinfection (for layer farms), the presence of the house hygiene locks (for broiler and layer farms), the cleanliness of the farm hygiene lock (for broiler farms), proximity to water sources (for turkey farms), and adequate coverage of built-up litter (for broiler and turkey farms). Further studies are needed to investigate the potential impact of biosecurity compliance on poultry flock health and disease status, as well as antimicrobial use as a consequence of the treatment of infectious diseases in poultry flocks.

## Figures and Tables

**Figure 1 animals-12-01409-f001:**
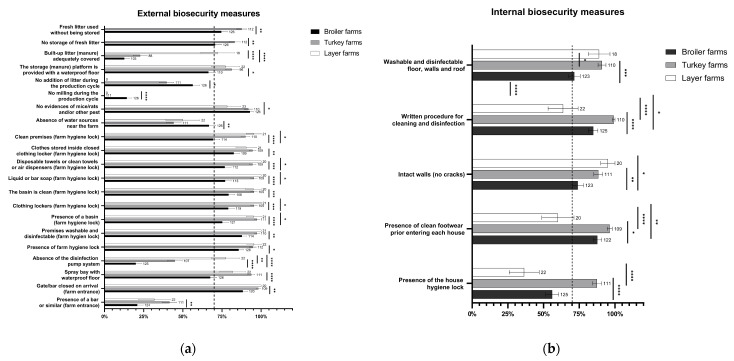
Level of significance among biosecurity measures in broiler, turkey, and layer farms. The figure shows the significance found between each biosecurity measure. (**a**) External biosecurity. (**b**) Internal biosecurity. The number of responding farms is reported beside each bar. *p* < 0.05 shown as *, *p* < 0.01 as **, *p* < 0.005 as ***, and *p* < 0.001 as ****. The whiskers represent the 95% confidence intervals (CI).

**Figure 2 animals-12-01409-f002:**
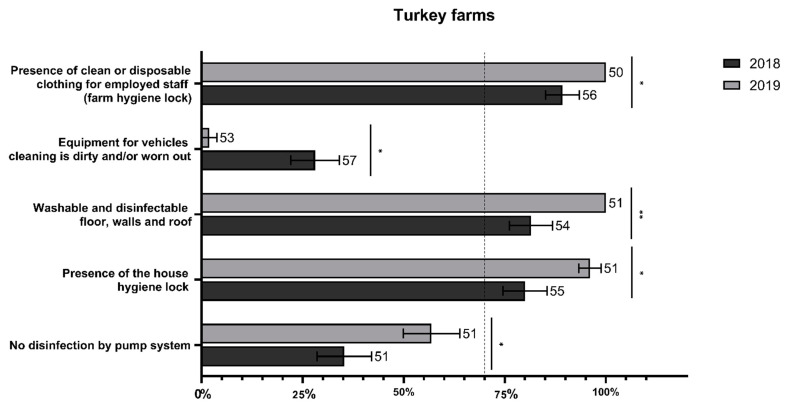
Level of significance among biosecurity measures in turkey farms during 2018 and 2019. The figure shows the significance found between biosecurity measures in turkey farms in the two-year period. The number of responding farms is reported beside each bar. *p* < 0.05 shown as *, *p* < 0.01 as **. The whiskers represent the 95% CIs.

**Table 1 animals-12-01409-t001:** Poultry farms investigated in 2018 and 2019 for biosecurity compliance. Farms are grouped according to the year of visit (2018 and 2019). The total number of farms visited in the two-year period is also reported.

	2018		2019		Total Amount of Farms Visited in 2018–2019 ^1^	
Farms	Number	Percentage (%)	Number	Percentage (%)	Number	Percentage (%)
Broiler farms	120	63.8	6	8.5	126	48.6
Turkey farms	58	30.9	53	74.6	111	42.9
Layer Farms	10	5.3	12	16.9	22	8.5
Total	188	100.0	71	100.0	259	100.0

^1^ A total of 3, 48, and 1 broiler, turkey, and layer farms, respectively, were visited twice between 2018 and 2019.

## Data Availability

Some of the data presented in this study are available upon request from the corresponding author. The data are not publicly available due to privacy concerns.

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
