# Peer review of "Assessing Biosecurity Compliance in Poultry Farms: A Survey in a Densely Populated Poultry Area in North East Italy"

_animals, 2022, doi:10.3390/ani12111409_

Round 1

Reviewer 1 Report

Dear authors,

Thank you for submitting such an interesting paper. The paper documents the assessment of biosecurity measures in poultry farms in North-East Italy. The paper is easy to read and to understand and it is well structured. I have very little comments or suggestions.

Please see below further comments and suggestions.

Kind regards

Simple summary and abstract

L21- insert “farms” or similar here: “with some …. still”

L32 - were the most compliant biosecurity >> were the biosecurity measures most complied to.

Introduction

Are there quality assurance schemes imposing stricter biosecurity measures than those mandatory by law?

Material and methods

Is there a website where the full check list used can be consulted? Are there annual reports on the biosecurity checks made by the competent authorities? Are they published somewhere?

How was data collected? In paper or using software? Was excel used to process and analyse data?

Results

Figure 1. Clean footwear seems to be less frequent in layers. Why is this? Layers are the ones staying there longer, it should be even more important to have clean footwear compared to broilers and turkeys.

All farms surveyed were indoor farms, correct?

Are there outdoor poultry production in the area? Is the biosecurity checklist similar for those?

Are there differences between the different vertically integrated systems? (do you have data on that?)

Discussion

L271 – the term used most often is antimicrobial use/usage (AMU). I leave it to you to decide whether you want to use AMU or AMC.

Author Response

Response to Reviewer 1 Comments

Dear authors,

Thank you for submitting such an interesting paper. The paper documents the assessment of biosecurity measures in poultry farms in North-East Italy. The paper is easy to read and to understand and it is well structured. I have very little comments or suggestions.

Please see below further comments and suggestions.

Kind regards

The Authors wish to thank the reviewer who appreciated the study and provided valuable comments for improving the manuscript.

Simple summary and abstract

Point 1: L21- insert “farms” or similar here: “with some …. still”

Revised as suggested; “measures” has been added to the text.

Point 2: L32 - were the most compliant biosecurity >> were the biosecurity measures most complied to.

Revised as suggested.

Introduction          

Point 3: Are there quality assurance schemes imposing stricter biosecurity measures than those mandatory by law?

The Authors are aware that integrated poultry companies adopt specific policies on biosecurity measures to implement in their farms, but they are confidential and unfortunately the Authors were not allowed to gain any knowledge about them. 

Material and methods

Point 4: Is there a website where the full check list used can be consulted? Are there annual reports on the biosecurity checks made by the competent authorities? Are they published somewhere?

Yes, the blank checklists can be found online but the filled ones are stored in a database hold by the official veterinary services and this database is not publicly available. Similarly, a report is produced annually but it is not publicly available.  

Point 5: How was data collected? In paper or using software? Was excel used to process and analyse data?

The Italian biosecurity assessment system described in the manuscript includes a compulsory in-person farm visit made by official veterinarians of the official veterinary services. Referring to this study, all the questionnaires were filled in by the official veterinarian on a paper copy of the checklists during the in-person visit of the farms. Afterward, the answers to each question were transferred into an excel file and analysed statistically by using GraphPad Prism.

Results

Point 6: Figure 1. Clean footwear seems to be less frequent in layers. Why is this? Layers are the ones staying there longer, it should be even more important to have clean footwear compared to broilers and turkeys.

The Authors thank the reviewer for the remark. Indeed, layer farmers should pay more attention to the cleaningless of footwear. A sentence regarding this non-compliance has been added to the discussion (lines 249-251).

Point 7: All farms surveyed were indoor farms, correct? Are there outdoor poultry production in the area? Is the biosecurity checklist similar for those?

Most of the surveyed farms were indoor, only a small amount of them (n = 13, n = 4, and n = 1 for broiler, turkey and layer farms, respectively) was outdoor. However, the checklist is the same for both. If the reviewer believes that reporting this information would be useful for the readers, the Authors are willing to comply with his/her request.

Point 8: Are there differences between the different vertically integrated systems? (do you have data on that?)

As described in the previous comment, integrated poultry companies keep their data confidential, and thus the Authors were not able to assess differences among the different companies. 

Discussion

Point 9: L271 – the term used most often is antimicrobial use/usage (AMU). I leave it to you to decide whether you want to use AMU or AMC.

Revised as suggested.

Reviewer 2 Report

Well done, nice paper ! and indeed interesting results that a worth to be published.

I have a few small items to be dealt with:

  • Line 82-84: from your manuscript it is not clear how many famrs are totally present in the region; please indicate those numbers for the different sectors. Did you investigate a sample of active farms, or all of them ?
  • Figure 1: you use error bars; in some of them the wisker of the upper-bound of the 95% confidence interval is above 100%, that should not be possible, so you have to have a look at that.  The question is furthermore: why do you have a confidence bound in your estimates ? I think because you have used a sample of farms in your investigation, so this also goes back to my first remark.
  •  

Author Response

Response to Reviewer 2 Comments

Well done, nice paper ! and indeed interesting results that a worth to be published.

The Authors wish to thank the reviewer who appreciated the study and provided valuable comments for improving the manuscript.

I have a few small items to be dealt with:

Point 1: Line 82-84: from your manuscript it is not clear how many famrs are totally present in the region; please indicate those numbers for the different sectors. Did you investigate a sample of active farms, or all of them ?

The Italian biosecurity assessment system described in the manuscript includes a compulsory in-person farm visit that is made by official veterinarians with the purpose to assess the compliance to biosecurity measures in poultry farms as required by the law. The farms’ number and selection is established by the veterinary services on a yearly basis, according to the poultry category (e.g., turkeys, chickens, etc.) and the epidemiological situation (i.e., presence/absence of disease outbreaks). In addition, the official veterinarians select farms on the basis of their level of biosecurity implementation by prioritizing those in which non-compliance in biosecurity measures implementation was reported in the previous visits. Therefore, only a part of the total number of active farms present in the region was investigated. If the reviewer believes that the information on the total number of farms present in the region would be useful for the readers, the Authors are willing to comply with his/her request.  

Point 2: Figure 1: you use error bars; in some of them the wisker of the upper-bound of the 95% confidence interval is above 100%, that should not be possible, so you have to have a look at that. The question is furthermore: why do you have a confidence bound in your estimates ? I think because you have used a sample of farms in your investigation, so this also goes back to my first remark.

As per previous comment, we investigated only a part of the farms present in the area. The Authors thank the reviewer to point out the inaccuracy regarding the 95% CI in the pictures. Mistakes have been duly amended.

Reviewer 3 Report

Dear authors,

I have read your manuscript with care and find the scientific relevance high. However, there are some issues I want to address.

Introduction

The first sentences of the introduction are very negatively worded toward conventional poultry production. Although the content is not wrong, I would suggest rephrasing L42-45.

The authors remain very vague on the national regulation. As a reader, I want to know, what is obliged (expect the periodic inspections) and what are the consequences. Whether the farmers receive financial support and/or penalties when not implementing biosecurity measures, can highly influence the results and is interesting to know in order to compare the results with other (EU) countries. Also, it is only stated on line 234 that the regulation was established in 2005. I would prefer to have this information already in the introduction.

M&M

L84-85: are the number of farms in this study a representation of the poultry production in the area? I.e. are there indeed much fewer layer farms or was the lower number due to another reason?

L109-110: when is decided to give recommendations, and when do they give penalties?

L126-129: is this 70% based on previous results? Or was this the overall mean in this study?

Results

Fig 1: I was really confused by the figures. For one, the titles/categories do not correspond to the supplementary material. Which makes it confusing. Secondly, only part of the questions was included. I assume only the significant differences?
I have some difficulties with this. In my opinion, this gives a wrong impression of the results and this is not emphasized further in the manuscript. I would prefer all data to be included in the figures or at least make this selection much more clear.

L158 (and further): the different measures are not clearly divided into internal and external biosecurity. Therefore I would be careful with the differentiation into external and internal as I do not believe a lay audience is familiar enough with the terms to make the distinction always. A suggestion could be to color-code all measures into external/internal.

Fig2: again I have some difficulties with this figure. As indicated by the number of farms, all turkey farms are included in the figure. But this does not give you a lot of information as only 48 of 111 farms were visited twice. In my opinion, a better and more accurate figure would be to only include the "twin" farms. Or you should have corrected for the farms that are twice in the dataset. None of this is emphasized further in the manuscript. 

Discussion:

L202: entrance of people = external biosecurity

L231: please clarify that this relates to vehicles

L239: Each integration has additional biosecurity guidelines? If they have their own/different measures, I would assume there was more variation in between the farms.

L246: don't fully agree with this. A FARM hygiene lock is there to keep pathogens out. A HOUSE hygiene lock is intended to decrease the spread between different groups of animals. Hence this relates more to internal biosecurity

L267: do you know if other areas have lower biosecurity scores? And are outbreaks there more common? This would empower this hypothesis even further.

L270-276: I would include some of this information in the introduction already. In the intro, there is little emphasis on the beneficial effects of biosecurity, further than increased animal health and welfare. 

General comment on the discussion: as this data is collected nationally, it would be a nice addition to give a summary of the biosecurity situation in 2005 (start of the legislation) and now. And has this indeed led to a lower incidence of poultry diseases? 

Conclusion

L289: no social factors were included in the study. I would be careful to include increased awareness as a factor then. 

Author Response

Response to Reviewer 3 Comments

I have read your manuscript with care and find the scientific relevance high. However, there are some issues I want to address.

The Authors wish to thank the reviewer who appreciated the study and provided valuable comments for improving the manuscript.

Introduction

Point 1: The first sentences of the introduction are very negatively worded toward conventional poultry production. Although the content is not wrong, I would suggest rephrasing L42-45.

Thank you for the suggestion, the Authors have rephrased the sentence.

Point 2: The authors remain very vague on the national regulation. As a reader, I want to know, what is obliged (expect the periodic inspections) and what are the consequences.

Considering the large number of questions included in the questionnaire (more than 180 questions), the Authors decided to describe and discuss only a part of them in order to avoid a lengthy and tedious manuscript. Therefore, the focus was on data derived from the questions that could be meaningful from an epidemiological point of view and thus to the scientific community in general, while avoiding those that could be deemed as uninformative (e.g., number of silos or similar). In the revised manuscript, we have added an additional supplementary file (Table S1) in which a list of all the biosecurity measures included in the checklists is reported.

Point 3: Whether the farmers receive financial support and/or penalties when not implementing biosecurity measures, can highly influence the results and is interesting to know in order to compare the results with other (EU) countries.

Usually, when a non-compliance in biosecurity is detected during the first visit, only a recommendation to implement that measure is made by the official veterinary services. Afterward, when the non-compliance continues, a penalty consisting of a ticket (the amount depends on the non-compliance) is applied. No financial supports are provided neither by the government or the integrated company. The farmer is asked to pay on his/her own any intervention to comply with the law.

Point 4: Also, it is only stated on line 234 that the regulation was established in 2005. I would prefer to have this information already in the introduction.

A brief sentence has been added to the introduction (lines 76-77).

M&M

Point 5: L84-85: are the number of farms in this study a representation of the poultry production in the area? I.e. are there indeed much fewer layer farms or was the lower number due to another reason?

The Italian biosecurity assessment system described in the manuscript includes a compulsory in-person farm visit that is made by official veterinarians with the purpose to assess the compliance to biosecurity measures in poultry farms as required by the law. The farms’ number and selection is established by the veterinary services on a yearly basis, according to the poultry category (e.g., turkeys, chickens, etc.) and the epidemiological situation (i.e., presence/absence of disease outbreaks). In addition, the official veterinarians select farms on the basis of their level of biosecurity implementation by prioritizing those in which non-compliance in biosecurity measures implementation was reported in the previous visit.

Point 6: L109-110: when is decided to give recommendations, and when do they give penalties?

Please refer to the answer provided for Point 3. For example, a recommendation is given by the official veterinary service when the non-compliance regards a minor biosecurity measure (e.g., missing data in the registers, lacking of disposable clothes or shoes, etc.); a penalty is given when the non-compliance is repeated over time or the farmer does not comply with a biosecurity measure after multiple recommendations (e.g., restocking of the flock without respecting the downtime according to the official veterinary service’s advise).

Point 7: L126-129: is this 70% based on previous results? Or was this the overall mean in this study?

The cut-off value of 70% was not based on previous results, but rather was defined on the basis of the (left-skewed) frequency distribution of biosecurity compliance among all farms (see the graph below), as 70% represents the overall mean value and upper threshold of the relatively long distribution tail after which highly compliant farms are located.

For the figure, please see the attached word file

Results

Point 8: Fig 1: I was really confused by the figures. For one, the titles/categories do not correspond to the supplementary material. Which makes it confusing. Secondly, only part of the questions was included. I assume only the significant differences?

I have some difficulties with this. In my opinion, this gives a wrong impression of the results and this is not emphasized further in the manuscript. I would prefer all data to be included in the figures or at least make this selection much more clear.

The Authors thank the reviewer for the suggestion, both figures and supplementary material have been modified accordingly. Figure 1 has been splitted in two figures 1a and 1b, including all the external and internal biosecuriy measures showing statistically significant differences. It is Authors’ opinion that it might had been misguided reporting the outcome of each variable in the figures considering a) the large number of variables included in the analysis (figures difficult to read) and b) that several biosecurity measures were implemented in the vast majority of farms, as stated in the manuscript. Therefore, it came spontaneous the decision to include only variables statistically significant different among poultry categories in the figures. Data regarding the compliance of all biosecurity measures are reported in the supplementary material (Table S2). If the reviewer believes that reporting this information would be useful for the readers, the Authors are willing to comply with his/her request.

Point 9: L158 (and further): the different measures are not clearly divided into internal and external biosecurity. Therefore I would be careful with the differentiation into external and internal as I do not believe a lay audience is familiar enough with the terms to make the distinction always. A suggestion could be to color-code all measures into external/internal.

The Authors agree with the reviewer that it is not easy to divide biosecurity measures in external and internal; however, the Authors based their classification on the Biocheck.UGent (https://biocheckgent.com/en/about-biosecurity-poultry#about-biosecurity-in-poultry-production). According to the previous comment, figure 1 has been splitted in two Figures 1a and 1b, including all the external and internal biosecuriy measures showing statistically significant differences.

Point 10: Fig2: again I have some difficulties with this figure. As indicated by the number of farms, all turkey farms are included in the figure. But this does not give you a lot of information as only 48 of 111 farms were visited twice. In my opinion, a better and more accurate figure would be to only include the "twin" farms. Or you should have corrected for the farms that are twice in the dataset. None of this is emphasized further in the manuscript.

The Authors would like to point out that out of 111 checklists filled in for turkey farms in the two-year period, 96 referred to 48 turkey farms that were visited twice, or rather once in 2018 and once in 2019 (Table 1). Therefore, for each of the 48 farms the same checklist was filled in twice, once in 2018 and once in 2019. These data allowed the Authors to further analyse the compliance of biosecurity measures between the two years to verify any variation in biosecurity implementation over time.

Discussion:

Point 11: L202: entrance of people = external biosecurity

Thank you for the suggestion, we have modified the manuscript accordingly (lines 198-200).

Point 12: L231: please clarify that this relates to vehicles

Revised as suggested.

Point 13: L239: Each integration has additional biosecurity guidelines? If they have their own/different measures, I would assume there was more variation in between the farms.

The Authors are aware that integrated poultry companies adopt specific policies on biosecurity measures to implement in their farms, but they are confidential and unfortunately the Authors were not allowed to gain any knowledge about them.

Point 14: L246: don't fully agree with this. A FARM hygiene lock is there to keep pathogens out. A HOUSE hygiene lock is intended to decrease the spread between different groups of animals. Hence this relates more to internal biosecurity

The Authors thank the reviewer for the comment, we have revised the manuscript accordingly (lines 247-248).

Point 15: L267: do you know if other areas have lower biosecurity scores? And are outbreaks there more common? This would empower this hypothesis even further.

The Authors agree with the reviewer that this information would be of great value to understand the impact of biosecurity implementation on the occurrence of disease outbreaks. Except for avian influenza, only fragmented and not representative data of the past and actual outbreaks situation are available. Indeed, the Italian biosecurity assessment system does not foresee any evaluation of the presence of the diseases or pathogens in farms. The only purpose of on-farm inspections made by official veterinarians is to assess the compliance to biosecurity measures in poultry farms as required by the law.

Point 16: L270-276: I would include some of this information in the introduction already. In the intro, there is little emphasis on the beneficial effects of biosecurity, further than increased animal health and welfare.

As suggested by the reviewer, a sentence in the introduction has been added.

Point 17: General comment on the discussion: as this data is collected nationally, it would be a nice addition to give a summary of the biosecurity situation in 2005 (start of the legislation) and now. And has this indeed led to a lower incidence of poultry diseases?

Please refer to Point 15’s reply. Since the Authors recognize the importance to correlate biosecurity to flock health, the reviewer’s suggestion could be taken into account for future studies.

Conclusion

Point 18: L289: no social factors were included in the study. I would be careful to include increased awareness as a factor then.

The sentence has been deleted.

Round 2

Reviewer 3 Report

I believe the authors have made substantial improvements to the manuscript in a limited period of time. I agree that this revised version is suited for publication.